# SCALE-EQUIVARIANT STEERABLE NETWORKS

**Ivan Sosnovik**[*], **Michał Szmaja**[†], **Arnold Smeulders**
UvA-Bosch Delta Lab
University of Amsterdam

## ABSTRACT

The effectiveness of Convolutional Neural Networks (CNNs) has been substantially attributed to their built-in property of translation equivariance. However, CNNs do not have embedded mechanisms to handle other types of transformations. In this work, we pay attention to scale changes, which regularly appear in various tasks due to the changing distances between the objects and the camera. First, we introduce the general theory for building scale-equivariant convolutional networks with steerable filters. We develop scale-convolution and generalize other common blocks to be scale-equivariant. We demonstrate the computational efficiency and numerical stability of the proposed method. We compare the proposed models to the previously developed methods for scale equivariance and local scale invariance. We demonstrate state-of-the-art results on the MNIST-scale dataset and on the STL-10 dataset in the supervised learning setting.

## 1 INTRODUCTION

Scale transformations occur in many image and video analysis tasks. They are a natural consequence of the variable distances among objects, or between objects and the camera. Such transformations result in significant changes in the input space which are often difficult for models to handle appropriately without careful consideration. At a high level, there are two modeling paradigms which allow a model to deal with scale changes: models can be endowed with an internal notion of scale and transform their predictions accordingly, or instead, models can be designed to be specifically invariant to scale changes. In image classification, when scale changes are commonly a factor of 2, it is often sufficient to make class prediction independent of scale. However, in tasks such as image segmentation, visual tracking, or object detection, scale changes can reach factors of 10 or more. In these cases, it is intuitive that the ideal prediction should scale proportionally to the input. For example, the segmentation map of a nearby pedestrian should be easily converted to that of a distant person simply by downscaling.

Convolutional Neural Networks (CNNs) demonstrate state-of-the-art performance in a wide range of tasks. Yet, despite their built-in translation equivariance, they do not have a particular mechanism for dealing with scale changes. One way to make CNNs account for scale is to train them with data augmentation Barnard & Casasent (1991). This is, however, suitable only for global transformations. As an alternative, Henriques & Vedaldi (2017) and Tai et al. (2019) use the canonical coordinates of scale transformations to reduce scaling to well-studied translations. While these approaches do allow for scale equivariance, they consequently break translation equivariance.

Several attempts have thus been made to extend CNNs to both scale and translation symmetry simultaneously. Some works use input or filter resizing to account for scaling in deep layers Xu et al. (2014); Kanazawa et al. (2014). Such methods are suboptimal due to the time complexity of tensor resizing and the need for interpolation. In Ghosh & Gupta (2019) the authors pre-calculate filters defined on several scales to build scale-invariant networks, while ignoring the important case of scale equivariance. In contrast, Worrall & Welling (2019) employ the theory of semigroup equivariant networks with scale-space as an example; however, this method is only suitable for integer downscale factors and therefore limited.

---

[*]Correspondence to `i.sosnovik@uva.nl`
[†]Work was done when the author had an internship at the UvA-Bosch Delta Lab
Source code is available at `http://github.com/isosnovik/sesn`

In this paper we develop a theory of scale-equivariant networks. We demonstrate the concept of steerable filter parametrization which allows for scaling without the need for tensor resizing. Then we derive scale-equivariant convolution and demonstrate a fast algorithm for its implementation. Furthermore, we experiment to determine to what degree the mathematical properties actually hold true. Finally, we conduct a set of experiments comparing our model with other methods for scale equivariance and local scale invariance.

The proposed model has the following advantages compared to other scale-equivariant models:

1. It is equivariant to scale transformations with arbitrary discrete scale factors and is not limited to either integer scales or scales tailored by the image pixel grid.

2. It does not rely on any image resampling techniques during training, and therefore, produces deep scale-equivariant representations free of any interpolation artifacts.

3. The algorithm is based on the combination of tensor expansion and 2-dimensional convolution, and demonstrates the same computation time as the general CNN with a comparable filter bank.

## 2 PRELIMINARIES

Before we move into scale-equivariant mappings, we discuss some aspects of equivariance, scaling transformations, symmetry groups, and the functions defined on them. For simplicity, in this section, we consider only 1-dimensional functions. The generalization to higher-dimensional cases is straightforward.

**Equivariance** Let us consider some mapping $g$. It is equivariant under $L_\theta$ if and only if there exists $L'_\theta$ such that $g \circ L_\theta = L'_\theta \circ g$. In case $L'_\theta$ is the identity mapping, the function $g$ is invariant.

In this paper we consider scaling transformations. In order to guarantee the equivariance of the predictions to such transformations, and to improve the performance of the model, we seek to incorporate this property directly inside CNNs.

**Scaling** Given a function $f : \mathbb{R} \to \mathbb{R}$, a scale transformation is defined as follows:

$$L_s[f](x) = f(s^{-1}x), \quad \forall s > 0 \tag{1}$$

We refer to cases with $s > 1$ as *upscale* and to cases with $s < 1$ as *downscale*. If we convolve the downscaled function with an arbitrary filter $\psi$ and perform a simple change of variables inside the integral, we get the following property:

$$[L_s[f] \star \psi](x) = \int_\mathbb{R} L_s[f](x')\psi(x' - x)dx' = \int_\mathbb{R} f(s^{-1}x')\psi(x' - x)dx'$$
$$= s \int_\mathbb{R} f(s^{-1}x')\psi(s(s^{-1}x' - s^{-1}x))d(s^{-1}x') = sL_s[f \star L_{s^{-1}}[\psi]](x) \tag{2}$$

In other words, *convolution of the downscaled function with a filter can be expressed through a convolution of the function with the correspondingly upscaled filter where downscaling is performed afterwards*. Equation 2 shows us that the standard convolution is not scale-equivariant.

**Steerable Filters** In order to make computations simpler, we reparametrize $\psi_\sigma(x) = \sigma^{-1}\psi(\sigma^{-1}x)$, which has the following property:

$$L_{s^{-1}}[\psi_\sigma](x) = \psi_\sigma(sx) = s^{-1}\psi_{s^{-1}\sigma}(x) \tag{3}$$

It gives a shorter version of Equation 2:

$$L_s[f] \star \psi_\sigma = L_s[f \star \psi_{s^{-1}\sigma}] \tag{4}$$

We will refer to such a parameterization of filters as *Steerable Filters* because the scaling of these filters is the transformation of its parameters. Note that we may construct steerable filters from any

function. This has the important consequence that it does not restrict our approach. Rather it will make the analysis easier for discrete data. Moreover, note that any linear combination of steerable filters is still steerable.

**Scale-Translation Group** All possible scales form the scaling group $S$. Here we consider the discrete scale group, i.e. scales of the form $\ldots a^{-1}, a^{-1}, 1, a, a^2, \ldots$ with base $a$ as a parameter of our method. Analysis of this group by itself breaks the translation equivariance of CNNs. Thus we seek to incorporate scale and translation symmetries into CNNs, and, therefore consider the Scale-Translation Group $H$. It is a semidirect product of the scaling group $S$ and the group of translations $T \cong \mathbb{R}$. In other words: $H = \{(s, t) | s \in S, t \in T\}$. For multiplication of group elements, we have $(s_2, t_2) \cdot (s_1, t_1) = (s_2 s_1, s_2 t_1 + t_2)$ and for the inverse $(s_2, t_2)^{-1} \cdot (s_1, t_1) = (s_2^{-1} s_1, s_2^{-1}(t_1 - t_2))$. Additionally, for the corresponding scaling and translation transformations, we have $L_{st} = L_s L_t \neq L_t L_s$, which means that the order of the operations matters.

From now on, we will work with functions defined on groups, i.e. mappings $H \to \mathbb{R}$. Note, that simple function $f : \mathbb{R} \to \mathbb{R}$ may be considered as a function on $H$ with constant value along the $S$ axis. Therefore, Equation 4 holds true for functions on $H$ as well. One thing we should keep in mind is that when we apply $L_s$ to functions on $H$ and $\mathbb{R}$ we use different notations. For example $L_s[f](x') = f(s^{-1}x')$ and $L_s[f](s', t') = f((s, 0)^{-1}(s', t')) = f(s^{-1}s', s^{-1}t')$

**Group-Equivariant Convolution** Given group $G$ and two functions $f$ and $\psi$ defined on it, $G$-equivariant convolution is given by

$$[f \star_G \psi](g) = \int_G f(g') L_g[\psi](g') d\mu(g') = \int_G f(g')\psi(g^{-1}g')d\mu(g') \tag{5}$$

Here $\mu(g')$ is the Haar measure also known as invariant measure Folland (2016). For $T \cong \mathbb{R}$ we have $d\mu(g') = dg'$. For discrete groups, the Haar measure is the counting measure, and integration becomes a discrete sum. This formula tells us that the output of the convolution evaluated at point $g$ is the inner product between the function $f$ and the transformed filter $L_g[\psi]$.

## 3 SCALE-EQUIVARIANT MAPPINGS

Now we define the main building blocks of scale-equivariant models.

**Scale Convolution** In order to derive scale convolution, we start from group equivariant convolution with $G = H$. We first use the property of semidirect product of groups which splits the integral, then choose the appropriate Haar measures, and finally use the properties of steerable filters. Given the function $f(s, t)$ and a steerable filter $\psi_\sigma(s, t)$ defined on $H$, a scale convolution is given by:

$$
\begin{aligned}
[f \star_H \psi_\sigma](s, t) &= \int_S \int_T f(s', t') L_{st}[\psi_\sigma](s', t') d\mu(s') d\mu(t') \\
&= \sum_{s'} \int_T f(s', t')\psi_{s\sigma}(s^{-1}s', t' - t)dt' = \sum_{s'}[f(s', \cdot) \star \psi_{s\sigma}(s^{-1}s', \cdot)](t)
\end{aligned} \tag{6}
$$

And for the case of $C_{\text{in}}$ input and $C_{\text{out}}$ output channels we have:

$$[f \star_H \psi_\sigma]_m(s, t) = \sum_{n=1}^{C_{\text{in}}} \sum_{s'}[f_n(s', \cdot) \star \psi_{n,m,s\sigma}(s^{-1}s', \cdot)](t), \quad m = 1 \ldots C_{\text{out}} \tag{7}$$

The proof of the equivariance of this convolution to transformations from $H$ is given in Appendix A.

Kondor & Trivedi (2018) prove that a feed-forward neural network is equivariant to transformations from $G$ if and only if it is constructed from G-equivariant convolutional layers. Thus Equation 7 shows the most general form of scale-equivariant layers which allows for building scale-equivariant convolutional networks with such choice of $S$. We will refer to models using scale-equivariant layers with steerable filters as Scale-Equivariant Steerable Networks, or shortly *SESN*[1]

**Nonlinearities** In order to guarantee the equivariance of the network to scale transformations, we use scale equivariant nonlinearities. We are free to use simple point-wise nonlinearities. Indeed,

---

[1]pronounced 'season'

point-wise nonlinearities $\nu$, like ReLU, commute with scaling transformations:

$$
\begin{aligned}
[\nu \circ L_s[f]](s', x') &= \nu(L_s[f](s', x')) = \nu(f(s^{-1}s', s^{-1}x')) \\
&= \nu[f](s^{-1}s', s^{-1}x') = [L_s \circ \nu[f]](s', x')
\end{aligned}
\tag{8}
$$

**Pooling** Until now we did not discuss how to convert an equivariant mapping to invariant one. One way to do this is to calculate the invariant measure of the signal. In case of translation, such a measure could be the maximum value for example.

First, we propose the maximum scale projection defined as $f(s, x) \to \max_s f(s, x)$. This transformation projects the function $f$ from $H$ to $T$. Therefore, the representation stays equivariant to scaling, but loses all information about the scale itself.

Second, we are free to use spatial max-pooling with a moving window or global max pooling. Transformation $f(s, x) \to \max_x f(s, x)$ projects the function $f$ from $H$ to $S$. The obtained representation is invariant to scaling in spatial domain, however, it stores the information about scale.

Finally, we can combine both of these pooling mechanisms in any order. The obtained transformation produces a scale invariant function. It is useful to utilize this transformation closer to the end of the network, when the deep representation must be invariant to nuisance input variations, but already has very rich semantic meaning.

## 4    IMPLEMENTATION

In this paragraph we discuss an efficient implementation of Scale-Equivariant Steerable Networks. We illustrate all algorithms in Figure 1. For simplicity we assume that zero padding is applied when it is needed for both the spatial axes and the scale axis.

**Filter Basis** A direct implementation of Equation 7 is impossible due to several limitations. First, the infinite number of scales in $S$ calls for a discrete approximation. We truncate the scale group and limit ourselves to $N_S$ scales and use discrete translations instead of continuous ones. Training of SESN involves searching for the optimal filter in functional space which is a problem by itself. Rather than solving it directly, we choose a complete basis of $N_b$ steerable functions $\Psi = \{\psi_{s^{-1}\sigma, i}\}_{i=1}^{N_b}$ and represent convolutional filter as a linear combination of basis functions with trainable parameters $w = \{w_i\}_{i=1}^{N_b}$. In other words, we do the following substitution in Equation 7: $\psi_\sigma \to \kappa = \sum_i w_i \Psi_i$

In our experiments we use a basis of 2D Hermite polynomials with 2D Gaussian envelope, as it demonstrates good results. The basis is pre-calculated for all scales and fixed. For filters of size $V \times V$, the basis is stored as an array of shape $[N_b, S, V, V]$. See Appendix C for more details.

**Conv $T \to H$** If the input signal is just a function on $T$ with spatial size $U \times U$, stored as an array of shape $[C_{\text{in}}, U, U]$, then Equation 7 can be simplified. The summation over $S$ degenerates, and the final result can be written in the following form:

$$
\texttt{convTH}(f, w, \Psi) = \texttt{squeeze}(\texttt{conv2d}(f, \texttt{expand}(w \times \Psi)))
\tag{9}
$$

Here $w$ is an array of shape $[C_{\text{out}}, C_{\text{in}}, N_b]$. We compute filter $w \times \Psi$ of shape $[C_{\text{out}}, C_{\text{in}}, S, V, V]$ and expand it to shape $[C_{\text{out}}, C_{\text{in}}S, V, V]$. Then we use standard 2D convolution to produce the output with $C_{\text{out}}S$ channels and squeeze it to shape $[C_{\text{out}}, S, U, U]$. Note that the output can be viewed as a stack of feature maps, where all the features in each spatial position are vectors of $S$ components instead of being scalars as in standard CNNs.

**Conv $H \to H$** The function on $H$ has a scale axis and therefore there are two options for choosing weights of the convolutional filter. The filter may have just one scale and, therefore, does not capture the correlations between different scales of the input function; or, it may have a non-unitary extent $K_S$ in the scale axis and capture the correlation between $K_S$ neighboring scales. We refer to the second case as *interscale interaction*.

It the first case $w$ has shape $[C_{\text{out}}, C_{\text{in}}, N_b]$ and Equation 7 degenerates in the same way as before

$$
\texttt{convHH}(f, w, \Psi) = \texttt{squeeze}(\texttt{conv2d}(\texttt{expand}(f), \texttt{expand}(w \times \Psi)))
\tag{10}
$$

We expand $f$ to an array of shape $[C_{in}S, U, U]$ and expand $w \times \Psi$ to have shape $[C_{out}S, C_{in}S, V, V]$. The result of the convolution is then squeezed in the same way as before.

In the case of interscale interaction, $w$ has shape $[C_{out}, C_{in}, K_S, N_b]$. We iterate over all scales in interaction, shift $f$ for each scale, choose a corresponding part of $w$, and apply convHH to them. We sum the obtained $K_S$ results afterwards.

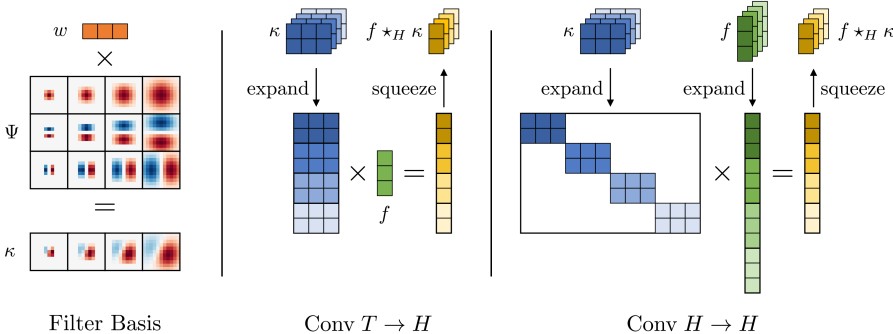

Filter Basis          Conv $T \rightarrow H$          Conv $H \rightarrow H$

Figure 1: Left: the way steerable filters are computed using steerable filter basis. Middle and right: a representation of scale-convolution using Equation 9 and Equation 10. As an example we use input signal $f$ with 3 channels. It has 1 scale on $T$ and 4 scales on $H$. It is convolved with filter $\kappa = w \times \Psi$ without scale interaction, which produces the output with 2 channels and 4 scales as well. Here we represent only channels of the signals and the filter. Spatial components are hidden for simplicity.

## 5   RELATED WORK

Various works on group-equivariant convolutional networks have been published recently. These works have considered roto-translation groups in 2D Cohen & Welling (2016a); Hoogeboom et al. (2018); Worrall et al. (2017); Weiler & Cesa (2019) and 3D Worrall & Brostow (2018); Kondor (2018); Thomas et al. (2018) and rotation equivariant networks in 3D Cohen et al. (2017); Esteves et al. (2018); Cohen et al. (2019). In Freeman & Adelson (1991) authors describe the algorithm for designing steerable filters for rotations. Rotation steerable filters are used in Cohen & Welling (2016b); Weiler et al. (2018a;b) for building equivariant networks. In Jacobsen et al. (2017) the authors build convolutional blocks locally equivariant to arbitrary $k$-parameter Lie group by using a steerable basis. And in Murugan et al. the authors discuss the approach for learning steerable filters from data. To date, the majority of papers on group equivariant networks have considered rotations in 2D and 3D, but have not payed attention to scale symmetry. As we have argued above, it is a fundamentally different case.

Many papers and even conferences have been dedicated to image scale-space — a concept where the image is analyzed together with all its downscaled versions. Initially introduced in Iijima (1959) and later developed by Witkin (1987); Perona & Malik (1990); Lindeberg (2013) scale space relies on the scale symmetry of images. The differential structure of the image Koenderink (1984) allows one to make a connection between image formation mechanisms and the space of solutions of the 2-dimensional heat equation, which significantly improved the image analysis models in the pre-deep learning era.

One of the first works on scale equivariance and local scale invariance in the framework of CNNs was proposed by Xu et al. (2014) named SiCNN. The authors describe the model with siamese CNNs, where the filters of each instance are rescaled using interpolation techniques. This is the simplest case of equivariance where no interaction between different scales is done in intermediate layers. In SI-ConvNet by Kanazawa et al. (2014) the original network is modified such that, in each layer, the input is first rescaled, then convolved and rescaled back to the original size. Finally, the response with the maximum values is chosen between the scales. Thus, the model is locally scale-invariant. In

| Method | Equivariance | Admissible Scales | Approach | Interscale |
|--------|:---:|:---:|:---:|:---:|
| SiCNN | ✓ | Grid | Filter Rescaling | ✗ |
| SI-ConvNet | ✗ | Grid | Input Rescaling | ✗ |
| SEVF | ✓ | Grid | Input Rescaling | ✓ |
| DSS | ✓ | Integer | Filter Dilation | ✓ |
| SS-CNN | ✗ | Any | Steerable Filters | ✗ |
| SESN, Ours | ✓ | Any | Steerable Filters | ✓ |

Table 1: Comparing SESN to SiCNN Xu et al. (2014), SI-ConvNet Kanazawa et al. (2014), SEVF Marcos et al. (2018), DSS Worrall & Welling (2019) and SS-CNN Ghosh & Gupta (2019). "Interscale" refers to the ability of capturing interscale interactions with kernels of non-unitary scale extent. "Grid" stands for the scales which generate images which lie exactly on the initial pixel grid.

Marcos et al. (2018), in the SEVF model, the input of the layers is rescaled and convolved multiple times to form vector features instead of scalar ones. The length of the vector in each position is the maximum magnitude of the convolution, while the direction of the angle encodes the scale of the image which gave this response. These scale-equivariant networks rely on image rescaling which is quite slow. Worrall & Welling (2019) (DSS) generalize the concept of scale-space to deep networks. They use filter dilation to analyze the images on different scales. While this approach is as fast as the standard CNN, it is restricted only to integer downscale factors $2, 4, 8 \ldots$. In Ghosh & Gupta (2019), while discussing SS-CNN the authors use scale-steerable filters to deal with scale changes. The paper does not discuss equivariance, which is an important aspect for scale.

We summarize the information about these models in Table 1. In contrast to other scale-equivariant models, SESN uses steerable filters which allows for fast scale-convolution with no limitation of flexibility. With the framework of Scale-Equivariant Convolutional Networks we are free to build both equivariant and invariant models of different kinds.

# 6 EXPERIMENTS

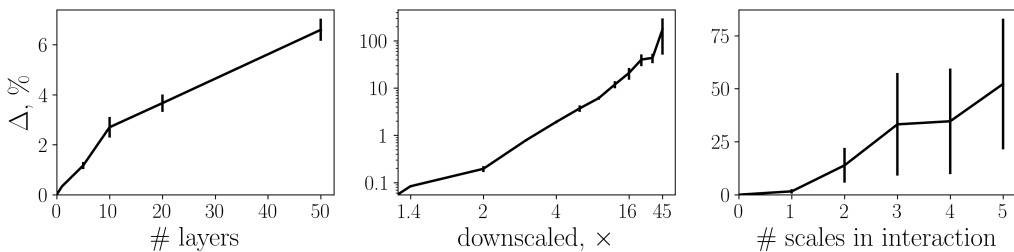

Figure 2: Equivariance error $\Delta$ as a function of the number of layers (left), downscaling applied to the input image (middle), and as a function of number of scales in interscale interactions (right). The bars indicate standard deviation.

In this section we conduct the experiments and compare various methods for working with scale variations in input data. Alongside SESN, we test local scale invariant SI-ConvNet and SS-CNN, scale equivariant SiCNN, SEVF and DSS. For SEVF, DSS and SS-CNN we use the code provided by authors, while for others we reimplement the main buildings blocks.

We provide additional experimental results on time performance of all these methods in Appendix B. Due to the algorithm proposed in Section 4 SESN allows for training several times faster than other methods which rely on image rescaling.

| Method | $(28 \times 28)$ | $(28 \times 28)$ + | $(56 \times 56)$ | $(56 \times 56)$ + | # Params |
|--------|-----------------|-------------------|------------------|-------------------|----------|
| CNN | $2.56 \pm 0.04$ | $1.96 \pm 0.07$ | $2.02 \pm 0.07$ | $1.60 \pm 0.09$ | 495 K |
| SiCNN | $2.40 \pm 0.03$ | $1.86 \pm 0.10$ | $2.02 \pm 0.14$ | $1.59 \pm 0.03$ | 497 K |
| SI-ConvNet | $2.40 \pm 0.12$ | $1.94 \pm 0.07$ | $1.82 \pm 0.11$ | $1.59 \pm 0.10$ | 495 K |
| SEVF Scalar | $2.30 \pm 0.06$ | $1.96 \pm 0.07$ | $1.87 \pm 0.09$ | $1.62 \pm 0.07$ | 494 K |
| SEVF Vector | $2.63 \pm 0.09$ | $2.23 \pm 0.09$ | $2.12 \pm 0.13$ | $1.81 \pm 0.09$ | 475 K |
| DSS Scalar | $2.53 \pm 0.10$ | $2.04 \pm 0.08$ | $1.92 \pm 0.08$ | $1.57 \pm 0.08$ | 494 K |
| DSS Vector | $2.58 \pm 0.11$ | $1.95 \pm 0.07$ | $1.97 \pm 0.08$ | $1.57 \pm 0.09$ | 494 K |
| SS-CNN | $2.32 \pm 0.15$ | $2.10 \pm 0.15$ | $1.84 \pm 0.10$ | $1.76 \pm 0.07$ | 494 K |
| SESN Scalar | $2.10 \pm 0.10$ | $1.79 \pm 0.09$ | $1.74 \pm 0.09$ | $1.50 \pm 0.07$ | 495 K |
| SESN Vector | $\mathbf{2.08 \pm 0.09}$ | $\mathbf{1.76 \pm 0.08}$ | $\mathbf{1.68 \pm 0.06}$ | $\mathbf{1.42 \pm 0.07}$ | 495 K |

Table 2: Classification error of different methods on MNIST-scale dataset, lower is better. In experiment we use image resolution of $28 \times 28$ and $56 \times 56$. We test both the regime without data augmentation, and the regime with scaling data augmentation, denoted with "+". All results are reported as mean $\pm$ std over 6 different fixed realizations of the dataset. The best results are **bold**.

## 6.1 EQUIVARIANCE ERROR

We have presented scale-convolution which is equivariant to scale transformation and translation for continuous signals. While translation equivariance holds true even for discretized signals and filters, scale equivariance may not be exact. Therefore, before starting any experiments, we check to which degree the predicted properties of scale-convolution hold true. We do so by measuring the difference $\Delta = \|[L_s\Phi(f) - \Phi L_s(f)\|_2^2 / \|L_s\Phi(f)\|_2^2$, where $\Phi$ is scale-convolution with randomly initialized weights.

In case of perfect equivariance the difference is equal to zero. We calculate the error on randomly sampled images from the STL-10 dataset Coates et al. (2011). The results are represented in Figure 2. The networks on the left and on the middle plots do not have interscale interactions. The networks on the middle and on the right plots consist of just one layer. We use $N_S = 5, 13, 5$ scales for the networks on the left, the middle, and the right plots respectively. While discretization introduces some error, it stays very low, and is not much higher than 6% for the networks with 50 layers. The difference, however, increases if the input image is downscaled more than 16 times. Therefore, we are free to use deep networks. However, we should pay extra attention to extreme cases where scale changes are of very big magnitude. These are quite rare but still appear in practice. Finally, we see that using SESN with interscale interaction introduces extra equivariance error due to the truncation of $S$. We will build the networks with either no scale interaction or interaction of 2 scales.

## 6.2 MNIST-SCALE

Following Kanazawa et al. (2014); Marcos et al. (2018); Ghosh & Gupta (2019) we conduct experiments on the MNIST-scale dataset. We rescale the images of the MNIST dataset LeCun et al. (1998) to $0.3 - 1.0$ of the original size and pad them with zeros to retain the initial resolution. The scaling factors are sampled uniformly and independently for each image. The obtained dataset is then split into 10,000 for training, 2,000 for evaluation and 50,000 for testing. We generate 6 different realizations and fix them for all experiments.

As a baseline model we use the model described in Ghosh & Gupta (2019), which currently holds the state-of-the-art result on this dataset. It consists of 3 convolutional and 2 fully-connected layers. Each layer has filters of size $7 \times 7$. We keep the number of trainable parameters almost the same for all tested methods. This is achieved by varying the number of channels. For scale equivariant models we add scale projection at the end of the convolutional block.

For SiCNN, DSS, SEVF and our model, we additionally train counterparts where after each convolution, an extra projection layer is inserted. Projection layers transform vector features in each spatial position of each channel into scalar ones. All of the layers have now scalar inputs instead of

vector inputs. Therefore, we denote these models with "Scalar". The original models are denoted as "Vector". The exact type of projection depends on the way the vector features are constructed. For SiCNN, DSS, and SESN, we use maximum pooling along the scale dimension, while for SEVF, it is a calculation of the $L_2$-norm of the vector.

All models are trained with the Adam optimizer Kingma & Ba (2014) for 60 epochs with a batch size of 128. Initial learning rate is set to 0.01 and divided by 10 after 20 and 40 epochs. We conduct the experiments with 4 different settings. Following the idea discussed in Ghosh & Gupta (2019), in addition to the standard setting we train the networks with input images upscaled to $56 \times 56$ using bilinear interpolation. This results in all image transformations performed by the network becoming more stable, which produces less interpolation artifacts. For both input sizes we conduct the experiments without data augmentation and with scaling augmentation, which results in 4 setups in total. We run the experiments on 6 different realizations of MNIST-scale and report mean $\pm$ std calculated over these runs.

The obtained results are summarized in Table 2. The reported errors may differ a bit from the ones in the original paper because of the variations in generated datasets and slightly different training procedure. Nevertheless, we try to keep our configuration as close as possible to Ghosh & Gupta (2019) which currently demonstrated the best classification accuracy on MNIST-scale. For example, SS-CNN reports error of $1.91 \pm 0.04$ in Ghosh & Gupta (2019) while it has $1.84 \pm 0.10$ in our experiments.

SESN significantly outperforms other methods in all 4 regimes. "Scalar" versions of it already outperform all previous methods, and "Vector" versions make the gain even more significant. The global architectures of all models are the same for all rows, which indicates that the way scale convolution is done plays an important role.

## 6.3 STL-10

In order to evaluate the role of scale equivariance in natural image classification, we conduct the experiments on STL-10 dataset Coates et al. (2011). This dataset consists of 8,000 training and 5,000 testing labeled images. Additionally, it includes 100,000 unlabeled images. The images have a resolution of $96 \times 96$ pixels and RGB channels. Labeled images belong to 10 classes such as bird, horse or car. We use only the labeled subset to demonstrate the performance of the models in the low data regime.

The dataset is normalized by subtracting the per-channel mean and dividing by the per-channel standard deviation. During training, we augment the dataset by applying 12 pixel zero padding and randomly cropping the images to size $96 \times 96$. Additionally, random horizontal flips with probability $50\%$ and Cutout DeVries & Taylor (2017) with 1 hole of 32 pixels are used.

As a baseline we choose WideResNet Zagoruyko & Komodakis (2016) with 16 layers and a widening factor of 8. We set dropout probability to 0.3 in all blocks. We train SESN-A with just vector features. For SESN-B we use maximum scalar projection several times in the intermediate layers, and for SESN-C we use interscale interaction.

| Method | Error, % | # Params |
|---|---|---|
| WRN | 11.48 | 11.0 M |
| SiCNN | 11.62 | 11.0 M |
| SI-ConvNet | 12.48 | 11.0 M |
| DSS | 11.28 | 11.0 M |
| SS-CNN | 25.47 | 10.8 M |
| SESN-A | 10.83 | 11.0 M |
| SESN-B | **8.51** | 11.0 M |
| SESN-C | 14.08 | 11.0 M |
| Harm WRN | 9.55 | 11.0 M |

Table 3: Classification error on STL-10. The best results are **bold**. We additionally report the current best result achieved by Harm WRN from Ulicny et al. (2019).

All models are trained for 1000 epochs with a batch size of 128. We use SGD optimizer with Nesterov momentum of 0.9 and weight decay of $5 \cdot 10^{-4}$. The initial learning rate is set to 0.1 and divided by 5 after 300, 400, 600 and 800 epochs.

The results are summarized in Table 3. We found SEVF training unstable and therefore do not include it in the table. Pure scale-invariant SI-ConvNet and SS-CNN demonstrate significantly worse results than the baseline. We note the importance of equivariance for deep networks. We also find that SESN-C performs significantly worse than SESN-A and SESN-B due to high equivariance

error caused by interscale interaction. SESN-B significantly improves the results of both WRN and DSS due to the projection between scales. The maximum scale projection makes the weights of the next layer to have a maximum receptive field in the space of scales. This is an easy yet effective method for capturing the correlations between different scales. This experiment shows that scale-equivariance is a very useful inductive bias for natural image classification with deep neural networks.

To the best of our knowledge, the proposed method achieves a new state-of-the-art result on the STL-10 dataset in the supervised learning setting. The previous lowest error is demonstrated in Ulicny et al. (2019). The authors propose Harm WRN — a network where the convolutional kernels are represented as a linear combination of Discrete Cosine Transform filters.

## 7 DISCUSSION

In this paper, we have presented the theory of Scale-Equivariant Steerable Networks. We started from the scaling transformation and its application to continuous functions. We have obtained the exact formula for scale-equivariant mappings and demonstrated how it can be implemented for discretized signals. We have demonstrated that this approach outperforms other methods for scale-equivariant and local scale-invariant CNNs. It demonstrated new state-of-the-art results on MNIST-scale and on the STL-10 dataset in the supervised learning setting.

We suppose that the most exciting possible application of SESN is in computer vision for autonomous vehicles. Rapidly changing distances between the objects cause significant scale variations which makes this well suited for our work. We especially highlight the direction of siamese visual tracking where the equivariance to principle transformations plays an important role.

ACKNOWLEDGMENTS

We thank Daniel Worrall for insightful discussion, Thomas Andy Keller, Victor Garcia, Artem Moskalev and Konrad Groh for valuable comments and feedback.

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

## A    PROOF OF EQUIVARIANCE

Let us first show that scale-convolution defined in Equation 6 is equivariant to translations.

$$
\begin{aligned}
[L_{\hat{t}}[f] \star_H \psi_\sigma](s,t) &= \sum_{s'} [L_{\hat{t}}[f](s', \cdot) \star \psi_{s\sigma}(s^{-1}s', \cdot)](t) \\
&= \sum_{s'} L_{\hat{t}}[f(s', \cdot) \star \psi_{s\sigma}(s^{-1}s', \cdot)](t) \\
&= L_{\hat{t}} \Big\{ \sum_{s'} [f(s', \cdot) \star \psi_{s\sigma}(s^{-1}s', \cdot)] \Big\}(t) \\
&= L_{\hat{t}}[f \star_H \psi_\sigma](s,t)
\end{aligned}
\tag{11}
$$

Now we show that scale convolution is equivariant to scale transformations:

$$
\begin{aligned}
[L_{\hat{s}}[f] \star_H \psi_\sigma](s,t) &= \sum_{s'} [L_{\hat{s}}[f](s', \cdot) \star \psi_{s\sigma}(s^{-1}s', \cdot)](t) \\
&= \sum_{s'} L_{\hat{s}}[f(\hat{s}^{-1}s', \cdot) \star \psi_{\hat{s}^{-1}s\sigma}(s^{-1}s', \cdot)](t) \\
&= \sum_{s''} [f(s'', \cdot) \star \psi_{\hat{s}^{-1}s\sigma}(\hat{s}s^{-1}s'', \cdot)](\hat{s}^{-1}t) \\
&= [f \star_H \psi_\sigma](\hat{s}^{-1}s, \hat{s}^{-1}t) \\
&= L_{\hat{s}}[f \star_H \psi_\sigma](s,t)
\end{aligned}
\tag{12}
$$

Finally, we can use the property of semidirect product of groups

$$
L_{\hat{s}\hat{t}}[f] \star_H \psi_\sigma = L_{\hat{s}}L_{\hat{t}}[f] \star_H \psi_\sigma = L_{\hat{s}}[L_{\hat{t}}[f] \star_H \psi_\sigma] = L_{\hat{s}}L_{\hat{t}}[f \star_H \psi_\sigma] = L_{\hat{s}\hat{t}}[f \star_H \psi_\sigma]
\tag{13}
$$

## B    TIME PERFORMANCE

We report the average time per epoch of different methods for scale equivariance and local scale invariance in Table 4. Experimental setups from Section 6.2 are used. We used 1 Nvidia GeForce GTX 1080Ti GPU for training the models.

The methods relying on image rescaling techniques during training (SiCNN, SI-ConvNet, SEVF) demonstrate significantly worse time performance that the ones, using either steerable filters or filter dilation. Additionally, we see that our method outperforms SS-CNN by a wide margin. Despite the similar filter sizes and comparable number of parameters between SS-CNN and SESN Scalar, the second one demonstrates significantly better results due to the algorithm proposed in Section 4. Finally, DSS performs slightly faster in some cases than our method as each convolution involves less FLOPs. Dilated filters are sparse, while steerable filters are dense.

| Method | $28 \times 28$, s | $56 \times 56$, s |
|---|---|---|
| CNN | 3.8 | 3.8 |
| SiCNN Scalar | 13.5 | 18.9 |
| SiCNN Vector | 15.3 | 22.8 |
| SI-ConvNet | 18.4 | 33.1 |
| SEVF Scalar | 21.0 | 38.4 |
| SEVF Vector | 25.4 | 46.0 |
| DSS Scalar | 3.9 | 5.0 |
| DSS Vector | 3.9 | 4.8 |
| SS-CNN | 14.8 | 16.6 |
| SESN Scalar | 3.8 | 5.1 |
| SESN Vector | 3.8 | 6.8 |

Table 4: Average time per epoch during training on input data with resolution $28 \times 28$ and $56 \times 56$.

## C    BASIS

Assuming that the center of the filter is point $(0, 0)$ in coordinates $(x, y)$, we use the filters of the following form:

$$\psi_\sigma(x, y) = A \frac{1}{\sigma^2} H_n\left(\frac{x}{\sigma}\right) H_m\left(\frac{y}{\sigma}\right) \exp\left[-\frac{x^2 + y^2}{2\sigma^2}\right] \tag{14}$$

Here $A$ is a constant independent on $\sigma$, $H_n$ — Hermite polynomial of the $n$-th order. We iterate over increasing pairs of $n, m$ to generate the required number of functions.

# D    MODEL CONFIGURATION

## D.1    MNIST-SCALE

| Method | Conv 1 | Conv 2 | Conv 3 | FC 1 | # Scales |
|---|---|---|---|---|---|
| CNN | 32 | 63 | 95 | | 1 |
| SiCNN | 32 | 63 | 95 | | 7 |
| SI-ConvNet | 32 | 63 | 95 | | 7 |
| SEVF Scalar | 32 | 63 | 95 | 256 | 8 |
| SEVF Vector | 23 | 45 | 68 | | 8 |
| DSS | 32 | 63 | 95 | | 4 |
| SS-CNN | 30 | 60 | 90 | | 6 |
| SESN | 32 | 63 | 95 | | 4 |

Table 5: Number of channels in convolutional layers, number of units in fully-connected layers and number of scales used by different models in Section 6.2.

## D.2    STL-10

| Method | Block 1 | Block 2 | Block 3 | # Scales |
|---|---|---|---|---|
| CNN | 16 | 32 | 64 | 1 |
| SiCNN | 16 | 32 | 64 | 3 |
| SI-ConvNet | 16 | 32 | 64 | 3 |
| SEVF | 11 | 23 | 45 | 3 |
| DSS | 16 | 32 | 64 | 4 |
| SS-CNN | 11 | 22 | 44 | 3 |
| SESN | 16 | 32 | 64 | 3 |

Table 6: Number of channels in convolutional blocks and number of scales used by different models in Section 6.2. We report the number of channels up to the widening factor.

