# OpenReview forum: "Scale-Equivariant Steerable Networks"
_ICLR.cc/2020/Conference — Accept (Poster)_

### Official Review · AnonReviewer2 · 2019-10-21
**Official Blind Review #2**

**Rating:** 8

**Review:**

This paper proposed scale-equivariant steerable convolutional neural networks that is able to preserve both the translation and scaling symmetry of the data in the representation. To achieve this, the authors developed the scale-convolution blocks in the network, and generalized other common blocks, such as pooling and nonlinearity, to remain scale-equivariant. Extensive experiments have been conducted to show that the proposed scale-equivariant network
(a) is indeed scale-equivariant even with numerical discretization
(b) achieves better classification performance when compared to non-scale-equivariant networks as well as previously proposed locally-scale invariant networks.

Overall, this is a very good paper. The paper is well-written and well-organized. The newly proposed scale-convolution is the most general way of achieving scale-equivariant representations. Experiments are convincing and justifies the usage of the proposed architecture in dealing with multi-scale inputs.

One question to ask that does not effect my rating:
There is a very similar paper submitted to this conference:
https://openreview.net/forum?id=rkgCJ64tDB
Would you care to make a comparison between these two manuscript?

**Experience Assessment:**

I have published in this field for several years.

**Review Assessment: Checking Correctness Of Derivations And Theory:**

I assessed the sensibility of the derivations and theory.

**Review Assessment: Checking Correctness Of Experiments:**

I assessed the sensibility of the experiments.

**Review Assessment: Thoroughness In Paper Reading:**

I read the paper at least twice and used my best judgement in assessing the paper.

---

> ### Author Response · Authors · 2019-11-14
> **Reply to Review #2**
>
> Thank you for your review.
>
> Q: Would you care to make a comparison between these two manuscripts?
>
> A: In “Scale-Equivariant Neural Networks with Decomposed Convolutional Filters” the authors propose scale-translation equivariant convolutional layers which are similar to what we propose. In addition, we propose the maximum scale projection which transforms the functions on scale-translations to the functions on just translations. In ScDCFNet paper this step is left unspecified. This difference has consequences for the experimental outcomes.

---

### Official Review · AnonReviewer3 · 2019-10-22
**Official Blind Review #3**

**Rating:** 6

**Review:**

The paper describes a method for integrating scale equivariance into convolutional networks using steerable filters.  After developing the theory using continuous scale and translation space, a discretized implementation using a fixed set of steerable basis elements is described.  Experiments are performed measuring the error from true equivariance, varying number of layers, image scale and scales in scale interactions.  The method is evaluated using MNIST-scale and STL-10, with convincing results on MNIST-scale and bit less convincing but still good results on STL-10.

Overall, I think this is a nice paper with generally good explanations and experiments probing the behavior.  I would have liked to see more probing into the effects of number and distance between scales.  Table 1 and corresponding text say that a significant advantage of the approach is that it can handle arbitrary scale values, but there was no explicit exploration of the effects of using this beyond one set of scales per experiment/dataset.  What scale values can be sampled, which work best, and why?

Also, while the MNIST-scale experiment seems convincing, I think the STL-10 is a bit less (but still OK):  Although the method outperforms other methods and appropriate baseline models, it's a little disappointing that pooling over scales (which I would would convert the equivariance to invariance) is best, and inter-scale interactions increase error.  (Perhaps this is not too surprising in retrospect, as images may have limited scale variation from camera position in this dataset, but significant within-class viewpoint variation.)

Even so, I still find the method concise and of interest, with the basics evaluated, even if some of its unique advantages may have been better explored.


Additional Questions:

* Inter-scale interaction could be elaborated a bit more.  End of sec 4 says, "use convHH for each scale sequentially and .. sum".  I believe this is sequencing over scales in the kernel; explaining a bit better how this is implemented, including the shape of w in this case, would be helpful.

* Which scales were chosen for the fixed basis?  How large in spatial extent are the kernels in the basis elements, at each scale?

* In the implementation, what is the value of V (sampled 2d conv kernel size)?


**Experience Assessment:**

I have read many papers in this area.

**Review Assessment: Checking Correctness Of Derivations And Theory:**

I assessed the sensibility of the derivations and theory.

**Review Assessment: Checking Correctness Of Experiments:**

I assessed the sensibility of the experiments.

**Review Assessment: Thoroughness In Paper Reading:**

I read the paper at least twice and used my best judgement in assessing the paper.

---

> ### Author Response · Authors · 2019-11-14
> **Reply to Review #3**
>
> Thank you for your review.
>
> We compared our results on STL-10 to the current state-of-the-art model in the supervised learning setting known as Harm-WRN. Our model SESN-B ourperformes it by more than 1% and achieves new state-of-the-art result on this dataset in the supervised learning setting. We include Harm-WRN  in Table 3 for comparison.
>
> Q: Inter-scale interaction could be elaborated a bit more. I believe this is sequencing over scales in the kernel.
>
> A: Indeed. You are right. In the case of interscale interaction $w$ has shape $[C_\text{out}, C_\text{in}, K_S, N_b ]$. We iterate over all scales in interaction, we shift $f$ for each scale and choose a corresponding part of $w$ and apply convHH. We sum the obtained results afterwards. Thank you for this comment. We rephrased this section of our paper to make it easier for understanding.
>
> -------------------------
>
> Q: Which scales were chosen for the fixed basis? How large in spatial extent are the kernels in the basis elements, at each scale? In the implementation, what is the value of V?
>
> A: The scales and therefore $\sigma$ are the hyperparameters of the proposed method. The set of values we choose from is tailored by the requirement of the completeness of the obtained basis on the smallest scale when it is projected to the pixel grid.
>
> For MNIST-scale experiment we used 4 scales with a step of $q=(10/3)^{1/3} \approx 1.49$. We generate filters for $\sigma=1.5, 1.5 q, 1.5 q^2, 1.5 q^3$ and store them in an array of spatial extent of $V=13$. We choose $q$ by relying on prior knowledge about the dataset. And the value of 1.5 is chosen from a set of $[1.1 - 1.7]$ with a step of 0.1 by using cross-validation. We choose the value which gives the best accuracy on the validation set. The variation of the accuracy during cross-validation is of 0.1% on the scale of about 2%.
>
> For STL-10 experiment, we sample 3 bases for $\sigma = 0.9, 0.9 \sqrt{2}, 1.8$ and store them in an array of spatial extent of $V= 7$. We chose the maximum number of scales we are able to use on our hardware. Here we use value of $0.9$ as it generated the complete basis on the smallest scale. And the value of $\sqrt{2}$ is motivated by the assumption that in natural images of cats, cars, horses, etc. the scale variations are usually of factor 2. We did not run cross validation on this dataset.

---

### Official Review · AnonReviewer1 · 2019-10-24
**Official Blind Review #1**

**Rating:** 6

**Review:**

This paper proposes a framework (SESN) for learning deep networks that possess scale equivariance in addition to translation invariance. The formulation is based on group convolution on the scale-translation group. Filters are represented as the coefficients of a set of continuous basis functions, which are sampled (once) at a discrete set of scales. The theoretical formulatioin is clear and interesting. The approach is evaluated in terms of image classification accuracy. The set of baselines is quite exhaustive, including recent papers and papers that are not widely-known.

The most significant improvement for the STL-10 dataset was obtained by the SESN-B variant. This is interesting, because it applies the same operation independently at multiple scales and periodically performs global pooling over scale.

The effectiveness of the approach was demonstrated in the low-data regime, where the inductive bias of scale equivariance is more likely to help.

Overall I found the paper to be thought-provoking and well-executed. There are a number of questions that I would still like to see investigated, but nevertheless I feel that this paper already represents a worthwhile contribution.

Most important issues and questions:

(1.1) The SESN-B architecture resembles quite closely the SI-ConvNet architecture of Kanazawa et al. (except that that paper resized the images instead of the filters). While your approach may be more computationally efficient, it's not clear what leads to the improvement in accuracy here? Can you explain the difference?

(1.2) I would have preferred to see the approach demonstrated on a task which possesses scale equivariance, such as semantic segmentation.

(1.3) To argue in favour of the continuous basis, it would have been more convincing to compare against directly learning the filters at the highest resolution and obtaining the other filters by downsampling. This would not represent a runtime cost during inference.

(1.4a) It seems that SESN-C should contain SESN-A as a special case. However, SESN-C is worse than SESN-A. Do you have any idea whether this is due to optimization difficulty or over-fitting? Could you compare the training objectives?
(1.4b) It is stated that the scale equivariance of SESN-C is worse than SESN-A and -B. However, it should still be scale equivariant, except for boundary effects in scale? What is the parameter N_S in this experiment compared to the number of scales S? And the same question for the plot on the right in Figure 2.
(1.4c) How does SESN-C have the same number of parameters as SESN-A and SESN-B? I thought that more parameters would be required to compute interscale interaction. Was the number of channels reduced?

Issues with clarity:

(2.1) The explanation of equation 10 is not clear. In particular, the diagonal structure in Figure 1 is not stated anywhere in the text, it is simply explained as an expansion from [C_out, C_in, S, V, V] to [C_out S, C_in S, V, V].

(2.2) It's not immediately apparent how multiple applications of convHH are used to provide interscale interaction. I assume it is achieved by shifting f or psi in the scale dimension for each application of convHH, or equivalently by modifying the base-scale in the basis?

(2.3) The explanation of "scalar" and "vector" variants in the experimental section was not perfectly clear. It is stated that "all the layers have now scalar input instead of vector input." However, I understood that the max-reduction was only over the scale dimension, not the channel dimension, so that the inputs are still vectors? This is confusing as a reader.

(2.4) The expansion of the filters to a diagonal structure is described in the "implementation" section. However, it seems that this would entail wasteful multiplications by zero. Nevertheless, SESN is shown to be highly efficient in the appendix. Do you avoid these pointless operations in the actual implementation?

(2.5) It was not immediately obvious how $\psi_{\sigma}(s, t)$ was related to $\psi_{\sigma}(x)$.

Other details:

(3.1) In Figure 2, how many layers does the network have which was used to construct the middle plot?

(3.2) It would have been useful to include a study of the effect of the range and resolution of the scale space.

**Experience Assessment:**

I have read many papers in this area.

**Review Assessment: Checking Correctness Of Derivations And Theory:**

I carefully checked the derivations and theory.

**Review Assessment: Checking Correctness Of Experiments:**

I carefully checked the experiments.

**Review Assessment: Thoroughness In Paper Reading:**

I read the paper thoroughly.

---

> ### Author Response · Authors · 2019-11-14
> **Reply to Review #1. Part 1**
>
> Thank you for your review.
>
>
> Q: (1.1) The SESN-B architecture resembles quite closely the SI-ConvNet architecture of Kanazawa et al. Can you explain the difference?
> A: SI-ConvNet uses image resizing in each convolutional layer of the network. It relies on the interpolation techniques which cause interpolation artifacts and lead to less stable optimization and as a results to a decreased classification accuracy.
>
> Q: (1.2) I would have preferred to see the approach demonstrated on a task which possesses scale equivariance, such as semantic segmentation.
> A: You are right. We are currently working on applying SESN to visual tracking, where scale varies frequently because of the rapidly changing distances between objects and a camera.
>
> Q: (1.3) It would have been more convincing to compare against directly learning the filters at the highest resolution and obtaining the other filters by downsampling.
> A: The approach you describe seems to be close to SiCNN by Xu et al. We compare SESN against this one in our paper and favorably so.
>
> Q: (1.4 a, b, c) It seems that SESN-C should contain SESN-A as a special case. However, SESN-C is worse than SESN-A.
> A: SESN-C has the same number of parameters as SESN-A and SESN-B. Indeed, it is achieved by reducing the number of channels of the model. A wider version of SESN-C with the same number of channels as in SESN-A would have about 12 M parameters and would contain SESN-A as a special case with some weights equal to zero. However, the wider SESN-C would pose a different optimization problem with respect to its weights.
>
> We used 3 scales for all SESN models in STL-10 experiment. SESN-C is scale-equivariant up to the border effects. This equivariance error, however, causes less generalization comparing to SESN-A and SESN-B.
>
> Q: (1.4 b) What is the number of scales in interaction in this experiment compared to the number of scales in S? And the same question for the plot on the right in Figure 2.
> A: In this experiment we used an interaction of 2 scales. And the total number of scales is equal to 3. In Figure 3, the total number of scales is 5 and the number of scales in interaction is represented on the horizontal axis.
>
> Q: (2.1, 2.4) The explanation of equation 10 is not clear
> A: In the proposed algorithm we represent convHH as a combination of tensor reshapings of different kinds and a convolution. In order to pack Equation 7 into a standard convolution, we expand a convolutional kernel to have a block-diagonal structure in the space of input-output channels. As a result, each scale of the kernel is convolved only with the function defined on the same scale. The obtained representation of the kernel is a sparse matrix. We delegate the further optimization of the proposed algorithm to a well-discovered problem of matrix multiplication. In our implementation we use a convolution routine of PyTorch. We are going to release our code soon.
>
> Q: (2.2) It's not immediately apparent how multiple applications of convHH are used to provide interscale interaction.
> A: In the case of interscale interaction $w$ has shape $[C_\text{out}, C_\text{in}, K_S, N_b ]$. we iterate over all scales in interaction, we shift $f$ for each scale and choose a corresponding part of $w$ and apply convHH. We sum the obtained results afterwards. Thank you for pointing this moment of our paper. We fixed it to make easier to understand.
>
> Q: (2.3) The explanation of "scalar" and "vector" variants in the experimental section was not perfectly clear.
> A: We rephrase this explanation in our text. We call some networks “vector” as they store $N_S$-dimensional vectors in each spatial position in each channel. In contrast, “scalar” networks or just standard CNNs have scalars in each position in each channel.
>
>
> Q: (2.5) It was not immediately obvious how $\psi(s, t)$ was related to $\psi(x)$.
> A: We modified it in the text to make it easier for understanding. $\psi(x) = \psi(s=1, t=x)$

---

> > ### Author Response · Authors · 2019-11-14
> > **Reply to Review #1. Part 2**
> >
> > Q: (3.1) In Figure 2, how many layers does the network have which was used to construct the middle plot?
> > A: In Figure 2 we represent the equivariance error of a network of just one layer. We added this information to Section 6.1 for better understanding.
> >
> > Q: (3.2) It would have been useful to include a study of the effect of the range and resolution of the scale space.
> > A: The scales and therefore $\sigma$ are the hyperparameters of the proposed method. The set of values we chose from is tailored by the requirement of the completeness of the obtained basis on the smallest scale when it is projected to the pixel grid.
> >
> > For MNIST-scale experiment we used 4 scales with a step of $q=(10/3)^{1/3} \approx 1.49$. We generate filters for $\sigma=1.5, 1.5 q, 1.5 q^2, 1.5 q^3$ and store them in an array of spatial extent of $V=13$. We choose $q$ by relying on prior knowledge about the dataset. And the value of 1.5 is chosen from a set of $[1.1 - 1.7]$ with a step of 0.1 by using cross-validation. We choose the value which gives the best accuracy on the validation set. The variation of the accuracy during validation is of 0.1% on the scale of about 2%.
> >
> > For STL-10 experiment, we sample 3 bases for $\sigma = 0.9, 0.9 \sqrt{2}, 1.8$ and store them in an array of spatial extent of $V= 7$. We chose the maximum number of scales we are able to use on our hardware. Here we use value of $0.9$ as it generated the complete basis on the smallest scale. And the value of $\sqrt{2}$ is motivated by the assumption that in natural images of cats, cars, horses, etc. the scale variations are usually of factor 2. We did not run cross validation on this dataset.

---

### Public Comment · ~K_V_Subrahmanyam1 · 2019-11-07
**A recent reference which also deals with scale.**

The following reference also deals with scale invariance using coupled-autoencoders. The approach is entirely different, but the idea is to deal with scales under rotations.

SO(2)-equivariance in Neural networks using tensor nonlinearity, Muthuvel Murugan, K V Subrahmanyam, BMVC 2019.

---

> ### Author Response · Authors · 2019-11-14
> **On the related paper**
>
> Thank you for this useful reference. We added it to our paper.

---

### Decision · Program_Chairs · 2019-12-19

**Decision:**

Accept (Poster)

**Comment:**

This work presents a theory for building scale-equivariant CNNs with steerable filters. The proposed method is compared with some of the related techniques . SOTA is achieved on MNIST-scale dataset and gains on STL-10 is demonstrated. The reviewers had some concern related to the method, clarity, and comparison with related works. The authors have successfully addressed most of these concerns. Overall, the reviewers are positive about this work and appreciate the generality of the presented theory and its good empirical performance. All the reviewers recommend accept.